# Finetuning Type I Interferon Signaling to Enhance T Cell Immunity in HIV Infection

**DOI:** 10.3390/v17060774

**Published:** 2025-05-29

**Authors:** Wenli Mu, Nandita Kedia, Anjie Zhen

**Affiliations:** Division of Hematology and Oncology, David Geffen School of Medicine, University of California, Los Angeles, CA 90095, USA

**Keywords:** type I interferon (IFN-I), human immunodeficiency virus (HIV), chronic virus infection, innate immunity, immune exhaustion, immune responses

## Abstract

Type I interferons (IFN-Is) have a complex role in HIV-1 infection, offering early protection but contributing to harm over time. In the acute phase, IFN-Is help control the virus, activate the innate immune system, and reduce the formation of long-term viral reservoirs. However, during chronic HIV-1 infection, continued IFN-I signaling can become harmful. It leads to ongoing inflammation, T cell exhaustion, and tissue damage. Prolonged IFN-I exposure also increases immune suppression by upregulating inhibitory receptors on T cells and disrupting their metabolism. New therapies are being developed to balance IFN-I’s benefits and drawbacks. These include blocking IFN-I receptors (IFNARs), enhancing autophagy, inhibiting JAK-STAT signaling, and combining immune checkpoint therapies. Such strategies aim to reduce immune dysfunction and improve T cell responses. Overall, understanding and adjusting IFN-I signaling could help manage chronic HIV-1 more effectively, minimizing harm while preserving its antiviral benefits.

## 1. Introduction

Human immunodeficiency virus (HIV) remains a major global health challenge, affecting over 38 million individuals worldwide [1,2]. Despite the substantial success of antiretroviral therapy (ART) in controlling viral replication and reducing mortality, immune dysfunction remains a major challenge in people living with HIV (PLWH) [3,4,5]. One of the critical drivers of this immune dysregulation is the dysregulated activation of type I interferon (IFN-I) signaling [6,7,8].

Type I IFNs, including IFN-α and IFN-β, are pivotal to the host’s antiviral defense, orchestrating early immune responses by stimulating the expression of interferon-stimulated genes (ISGs) [9,10,11,12]. The induction of type I IFN signaling occurs upon the recognition of specific ligands, such as viral nucleic acid or viral proteins by pattern recognition receptors (PRRs) such as Toll-like receptors (TLRs), RIG-I receptors (RLRs), NOD-like receptors (NLRs), etc., in response to viral infection [13]. The recognition of PRRs activates signaling pathways that lead to the production of IFN-I, which diffuses into the extracellular space. IFN-I binds to the IFN receptor (IFNAR) on the surface of nearby cells, including immune cells and epithelial cells [11]. The binding of IFN-I to IFNAR activates the JAK-STAT signaling pathway [11]. Upon binding, IFNAR1-associated protein tyrosine kinase 2 (Tyk2) and IFNAR2-associated protein tyrosine kinase 1 (JAK1) initiate phosphorylation events that activate a range of signal transducer and activator of transcription (STAT) proteins [11]. These activated STAT proteins then dimerize and translocate to the nucleus [11]. In the nucleus, phosphorylated STAT proteins bind to interferon-stimulated response elements (ISREs) and activate the transcription of ISGs [10]. ISGs encode a diverse set of proteins with antiviral functions, including the inhibition of viral entry, reverse transcription, and release; the degradation of viral RNA; and the modulation of host immune responses [9,12]. For example, IFITM (interferon-induced transmembrane) proteins block viral entry [14], ISG15 functions as a ubiquitin-like modifier that is conjugated to viral and host proteins, affecting their stability or localization [15], and OAS (2′-5′-oligoadenylate synthetase) activates RNase L, leading to the degradation of viral RNA [16]. Mx proteins are GTPases that trap viral nucleocapsids and prevent their nuclear import or replication [17]. These genes are essential for controlling viral replication during the acute phase of infection [18,19,20,21,22].

Despite its critical role in antiviral response, chronic type I IFN signaling is involved in driving chronic inflammation and immune exhaustion [23]. A defining feature of chronic HIV infection is the persistence of type I IFN responses, characterized by sustained ISG expression and prolonged immune activation [24,25,26,27,28]. In humans, the persistent activation of type I IFN pathways during chronic HIV infection leads to the upregulation of inhibitory receptors such as PD-1, TIM-3, and LAG-3 on T cells [29,30]. This contributes to immune exhaustion, characterized by diminished T cell proliferation, impaired cytokine production, and reduced cytotoxic activity [31,32,33,34]. Chronic IFN-I signaling also disrupts immune homeostasis by promoting systemic inflammation, tissue fibrosis, and metabolic dysregulation in key lymphoid compartments such as gut-associated lymphoid tissue (GALT) [33,35,36,37]. These pathological changes compromise immune restoration even in individuals receiving long-term ART. This review explores the multifaceted role of type I IFN signaling in HIV infection, with a particular focus on its dualistic impact on T cell immunity. While early IFN-I responses are critical for limiting viral replication and activating innate immunity, their prolonged activation contributes to immune suppression and disease progression. Emerging therapeutic strategies aimed at modulating type I IFN responses offer potential avenues to restore immune function, enhance T cell efficacy, and improve clinical outcomes in individuals living with HIV.

## 2. Early Protective Roles of Type I IFNs During Acute HIV Infection

Type I IFNs, including IFN-α and IFN-β, are critical cytokines that mediate the early host immune response to HIV infection [11,38]. During the acute phase of HIV infection, type I IFNs are rapidly produced by infected cells and plasmacytoid dendritic cells (pDCs) upon sensing viral RNA through PRRs, including endosomal TLR7 and such as cytosolic sensors (e.g., RLRs and cGAS-STING) [39,40]. This production of type I IFNs initiates a cascade of signaling pathways through the JAK-STAT pathway, leading to the induction of hundreds of ISGs [10,41]. These ISGs are responsible for targeting and inhibiting various stages of the HIV lifecycle, contributing to the early containment of the virus [42,43]. One well-studied ISG is tetherin (BST2), a protein that restricts viral particle release by anchoring budding virions to the host cell membrane, preventing their dissemination [42,44,45]. Research has shown that tetherin expression is upregulated in response to IFN-I signaling, effectively reducing the spread of HIV during early infection stages [44]. Similarly, APOBEC3G, an ISG and cytidine deaminase, induces hypermutation in the viral genome during reverse transcription in the absence of HIV viral protein Vif, rendering the virus replication incompetent [46,47,48]. Another critical ISG, SAMHD1, restricts viral replication by depleting the pool of deoxynucleotide triphosphates (dNTPs) required for reverse transcription. Studies have demonstrated that SAMHD1-mediated dNTP depletion, induced by type I IFNs, significantly limits HIV replication in macrophages and dendritic cells [49,50].

Type I IFNs are pivotal not only in initiating antiviral defenses but also in amplifying and coordinating the innate immune response. For example, they enhance the cytotoxic activity of natural killer (NK) cells, enabling them to target and kill infected cells more effectively [51,52]. Type I IFNs upregulate the expression of ligands for activating NK cell receptors, such as NKG2D, thereby increasing their susceptibility to NK-mediated killing during chronic viral infection [53,54]. Furthermore, IFN-Is stimulate pDCs, enhancing their antigen-presenting capacity and facilitating the activation of adaptive immune responses [55,56,57]. IFN-α has been shown to improve DC maturation and upregulate the expression of major histocompatibility complex (MHC) molecules, which are essential for T cell activation [58,59,60].

Robust type I IFN responses play a pivotal role in controlling acute HIV infection. Studies have shown that, during primary HIV infection, a high plasma viral load triggers a rapid but transient increase in multiple cytokines and chemokines, particularly type I IFNs [61]. This early IFN-I surge enhances antiviral immunity by stimulating the immune system and inducing antiviral factors, ultimately leading to lower viral loads. Individuals with a higher ISG expression in the early stages of HIV infection tend to exhibit lower viral loads and slower disease progression [62]. A clinical study by Stacey et al. on acute HIV patients monitored plasma viral load prior to infection and found that IFN-α and its mediators rise weeks before peak viremia, suggesting a predictive role in early immune responses [61]. Additionally, in asymptomatic HIV-infected individuals, IFN-α treatment has been shown to reduce viral isolation frequency and slow disease progression [63].

Collectively, these findings underscore the critical protective role of type I IFNs in acute HIV infection. However, the relationship between IFN levels and HIV disease progression is complex, influenced by factors such as specific IFN genes activated, downstream gene expression patterns, and the infection stage [31,64]. These findings highlight the critical role of type I IFNs in the host defense against HIV and their potential as therapeutic targets. Modulating IFN responses to enhance their protective effects while minimizing their detrimental consequences during chronic infection is an area of active research.

## 3. Chronic Dysregulation of Type I IFN Signaling

In contrast to its beneficial role in early infection, persistent IFN signaling during chronic HIV infection drives immune activation and dysfunction, leading to T cell and NK cell exhaustion and systemic inflammation. This chronic activation is fueled by low-level viral replication and microbial translocation from the gut, which continuously stimulates IFN-I production [65]. Non-human primate studies provided the first evidence linking excessive type I IFN responses to chronic immune activation. Natural hosts of Simian immunodeficiency virus (SIV), such as African green monkeys and sooty mangabeys, exhibit a controlled immune response where ISG expression spikes during acute infection but is quickly downregulated in the chronic phase, allowing them to avoid long-term immune activation [66,67,68]. In contrast, non-natural hosts like macaques experience prolonged ISG activation, which drives chronic inflammation and progression to AIDS [69,70,71,72]. These early findings demonstrated how an unregulated type I IFN response can transform an initially protective antiviral mechanism into a driver of immune dysfunction.

Similar patterns have been observed in humans. As HIV-1 infection progresses from the acute to the chronic phase, type I IFN signaling shifts from a protective antiviral response to a driver of immune pathology. Persistently elevated levels of ISGs are closely associated with CD4 T cell depletion and altered T cell dynamics [73,74]. For example, high levels of ISG15, an IFN-stimulated gene, were found to correlate with high viral loads and low CD4+ cell counts in patients with HIV [75]. The prolonged type I IFN responses in conjunction with sustained alterations in CD4+ and CD8+ T cell signaling pathways [76] exacerbate immune exhaustion and promote chronic inflammation—hallmarks of chronic HIV-1 infection [30,77,78,79].

Interestingly, several studies have reported that women exhibit a stronger IFN-I response during the acute phase of HIV infection, resulting in a lower viral reservoir compared to men [80]. One key reason for this difference is attributed to the increased capacity of pDCs from females to produce IFN-α upon TLR7 stimulation. During HIV-1 infection, TLR7 activation in pDCs drives robust IFN-I production. Notably, the TLR7 gene escapes X-chromosome inactivation in the immune cells of females, leading to higher TLR7 expression and stronger IFN-1 signaling compared to males [81,82]. Additionally, ISGs such as MX1, IFIT1, IFIT3, ISG15, OAS2, and IRF7 are more highly expressed in T cells and dendritic cells of HIV-infected women [83]. Despite this early advantage, women progress to AIDS faster than men at equivalent viral loads [84,85]. These findings underscore the dual role of IFN-I signaling in HIV infection—beneficial in early control but could be detrimental when chronically activated.

Persistent type I IFN signaling during chronic HIV infection induces the sustained expression of inhibitory receptors such as PD-1, TIM-3, and LAG-3 on CD4+ and CD8+ T cells [29,86,87], which impair T cell activation, proliferation, and effector functions, ultimately leading to immune exhaustion [26,88,89]. Prolonged IFN signaling also shifts cellular metabolism toward oxidative phosphorylation, reducing the glycolytic activity required for effective T cell proliferation and effector function [90,91]. This metabolic dysregulation contributes to the reduced functionality and survival of HIV-specific T cells, exacerbating immune exhaustion [90,91]. Persistent type I IFN also promotes the release of inflammatory cytokines, such as IL-10 and TGF-β, which are known to suppress T cell activity and reinforce an immunosuppressive environment [23,92,93,94]. Moreover, IFN-induced pro-apoptotic signals promote the premature death of activated T cells, diminishing the pool of functional HIV-specific T cells [95,96]. Prolonged type I IFN signaling creates a feedback loop of immune suppression and inflammation in chronic HIV infection, leading to T cell exhaustion, metabolic dysregulation, and cytokine-mediated suppression, all of which impair the immune system’s ability to control HIV replication and viral reservoirs [90].

Building on these systemic effects, recent studies have shown that type I IFN responses display tissue-specific dynamics, which also play a critical role in disease pathogenesis during chronic HIV infection. IFN-I, particularly IFNα, exhibit localized expression patterns during HIV-1 pathogenesis, with significantly higher expression in lymph node tissues compared to peripheral blood during chronic untreated infection. Despite ART, type I IFN signaling persists in gut tissues, likely driven by the early migration of pDCs—a key source of IFN-I—to mucosal and lymphoid sites during acute infection [97]. The sustained IFN-I activity fosters a chronic inflammatory environment that contributes to tissue fibrosis in lymphoid organs, including lymph nodes and GALT [98,99,100,101]. Fibrosis disrupts lymphoid architecture and impairs cell-to-cell interactions necessary for effective immune responses, leading to further CD4⁺ T cell depletion and immune dysfunction [98,99,102]. Persistent IFN-I signaling in GALT may exacerbate gut integrity and promote microbial translocation and further amplify systemic inflammation [103]. This chronic immune activation, driven in part by IFN-I, not only sustains viral reservoirs but also contributes to non-AIDS-related comorbidities, such as cardiovascular disease and metabolic disorders, even in individuals on suppressive ART [79,104]. Understanding these interactions between IFN-I, tissue microenvironments, and microbial factors could pave the way for novel interventions aimed at reducing inflammation and improving long-term outcomes in PLWH.

Primate studies further underscore the double-edged sword nature of type I IFN responses. Blocking IFNAR during acute SIV infection in rhesus macaques worsened outcomes by increasing reservoir size, decreasing antiviral gene expression, and accelerating disease progression, highlighting the importance of early IFN-I responses [69]. Conversely, prolonged IFNα administration led to desensitization, reduced ISG expression, and accelerated CD4+ T cell depletion, illustrating the harmful effects of sustained signaling [69]. However, blocking IFNAR in combination with ART during the chronic phase enhanced viral suppression, reduced reservoirs, and restored CD8+ T cell functions [105]. In sum, these findings suggested the complex role of type I IFNs in HIV-1 infection, highlighting that finetuning IFN-I responses may offer therapeutic benefits by balancing early immune activation with the prevention of long-term immune dysfunction.

## 4. Mechanisms of Immune Response Impairment by Type I IFNs

Type I IFNs significantly influence immune cell metabolism, particularly by inducing metabolic reprogramming to support immune activation. IFN-I helps promote glycolysis in immune cells, including macrophages and dendritic cells, enabling rapid and early innate immune responses to meet the increased energy demands during immune activation [91,106,107]. This metabolic shift allows immune cells to generate ATP quickly, ensuring that they have the energy necessary for their functions. For instance, Pantel et al. demonstrated that the direct activation of dendritic cells by IFN-I leads to a metabolic shift towards glycolysis, which is essential for their maturation and function [106]. Hedl et al. have shown that IRF5 can lead to increased glycolysis (a metabolic pathway) and promote the polarization of human macrophages towards an inflammatory M1 phenotype [107]. Additionally, IFN-I can modulate glucose metabolism through PI3K/Akt signaling, which has been shown to be essential for antiviral responses, as observed in the regulation of glucose metabolism during Coxsackievirus B3 infection [108].

However, excessive reliance on glycolysis can compromise mitochondrial function, leading to reduced oxidative phosphorylation and impaired energy production [109,110]. Lewis et al. showed that IFN-I inhibits mitochondrial activity, which may impair cellular energy production, and this has been observed in CD4+ T cells treated with IFN-β, which correlates with impaired mitochondrial function in diseases like multiple sclerosis [111,112]. Olson et al. have also shown that the induction of IFN-β correlates with impaired mitochondrial function, which potentially disrupts this balance between glycolysis and oxidative phosphorylation, critical for maintaining efficient cellular energy metabolism [113]. In juvenile systemic lupus erythematosus (SLE), a chronic autoimmune disease, type I IFN signaling has been associated with mitochondrial dysfunction in cytotoxic CD8+ T cells [114]. CD8+ T cells from IFN-high SLE patients exhibit enlarged mitochondria and reduced respiratory capacity, while exposure to IFNα increases NAD+ consumption. Prolonged IFNα exposure depletes NAD+, a key molecule for mitochondrial function, further impairing the energy production capacity of CD8+ T cells [114]. Additionally, type I IFNs elevate the production of reactive oxygen species (ROS) in T cells [114] and hematopoietic cells [115], further compromising mitochondrial integrity and function. Therefore, those studies highlighted the detrimental effects of sustained IFN-I activity on mitochondrial health.

Type I IFNs reshape the cytokine environment by promoting the production of pro-inflammatory cytokines such as IL-6, TNF-α, and IL-12 [61,116]. These cytokines indirectly support early antiviral defenses by enhancing Th1 polarization and increasing IFN-γ production through STAT4 activation [117,118,119]. Additionally, they promote the recruitment of cytotoxic CD8+ T cells via the TNF-α-induced upregulation of adhesion molecules like ICAM-1, facilitating an early immune response against HIV infection [120]. Furthermore, IFN-I provides a third signal to CD8+ T cells through a STAT4-dependent pathway, promoting their survival, cytolytic function, and IFN-γ production [121]. However, sustained IFN-I signaling in chronic infection triggers the excessive production of pro-inflammatory mediators (e.g., IL-6, CXCL10) while creating an immunosuppressive environment that hinders effective T cell responses. One of the primary ways is through the induction of anti-inflammatory cytokine IL-10 [122]. Interleukin-10 (IL-10) plays a crucial role in suppressing T cell responses and inducing immune tolerance. IL-10 directly inhibits T cell proliferation and cytokine production [123]. IL-10 suppresses T cell responses through the downregulation of APC co-stimulatory molecules (CD80/CD86) and the inhibition of DC-derived IL-12 (via SOCS3-mediated NF-κB suppression) [124]. In addition, TGF-β utilizes various mechanisms to inhibit the proliferation and reactivity of T lymphocytes by the suppression of IL-2 production, the downregulation of c-myc, and the suppression of T cell responses by regulating Foxp3+ Tregs [125,126,127,128]. Together, these cytokine-driven changes create a feedback loop of immune suppression that limits T cells’ ability to mount robust responses against viral replication. Altogether, this explains why the altered cytokine landscape is a key contributor to the inability to control HIV replication and clear latent reservoirs [129,130].

Beyond altering soluble mediators, type I IFNs indirectly impair immune responses by modulating antigen-presenting cells (APCs) themselves. Chronic IFN-I exposure skews monocytes and dendritic cells toward a dysregulated or suppressive phenotype [131]. In persistent viral infection models, IFN-I signaling drives the differentiation of “immunoregulatory” DCs that produce IL-10 and express inhibitory ligands such as PD-L1 while simultaneously blunting the generation of conventional DCs with T cell-stimulatory capacity [132]. These IFN-I conditioned APCs can attenuate T cell activation through multiple pathways. Notably, the upregulated PD-L1 on DCs and macrophages engages PD-1 on T cells, reinforcing T cell exhaustion and functional inactivation [132]. IFN-I signaling in DCs also induces immunosuppressive enzymes like indoleamine 2,3-dioxygenase (IDO), which catabolizes tryptophan and thereby stifles T cell proliferation as a negative feedback mechanism [133].

Concomitantly, persistent IFN-I-driven inflammation alters the architecture of lymphoid tissues, further impairing immune competence. During chronic HIV/SIV infection, lymph nodes and gut-associated lymphoid tissue undergo structural alterations, including fibrosis and the disorganization of tissue architecture within the organs [100,134,135]. Prolonged IFN-I and inflammatory cytokines (e.g., TNF-α) stimulate fibroblastic reticular cells and stromal cells within lymphoid tissues to produce excess collagen and other extracellular matrix proteins [136], leading to fibrotic scarring of the T cell zones [137]. This fibrosis, which is largely absent in natural SIV hosts that tightly regulate type I IFN responses, correlates with the collapse of the follicular dendritic cell network in germinal centers and a loss of normal lymphoid tissue organization [100,138]. The outcome is a lymphoid infrastructure less capable of supporting effective immune surveillance. The disruption of stromal scaffolds and chemokine gradients further limits T cell positioning and survival, thereby exacerbating the immune dysfunction driven by chronic IFN-I signaling.

Type I IFNs, in cooperation with TNF, induce the tandem occupancy of regulatory elements at non-ISGs that encode inflammatory mediators by IRFs and NF-κB, a process associated with increased histone marks that facilitate transcription, leading to faster and higher ISG production in response to subsequent environmental rechallenges [139,140]. When T cells are continuously exposed to a chronic inflammatory environment, they undergo extensive epigenetic changes, including DNA methylation, histone acetylation, and chromatin remodeling, resulting in long-lasting alterations to their gene expression profiles and functional states [141]. Studies have shown that exhaustion genes, including PD-1, LAG-3, and TIM-3, are epigenetically modulated by type I IFNs, potentially leading to a sustained expression of these inhibitory receptors [29]. The expressions of IFNs and ISGs are quantitatively controlled in a cell lineage-specific manner by H3K9me2, a negative histone marker. Weak producers of IFN (basal level) have significantly lower levels of H3K9me2 at IFNs and ISG promoters [142], while higher levels of H3K9 methylation lead to a decrease in inflammatory chemokine production [143]. Additionally, stimulation with type I IFNs increases the amount of trimethylated histone H3 Lys 4 (H3K4me3), a histone mark that promotes transcription at the promoters of genes encoding inflammatory mediators [141]. These data show the potential of the epigenetic landscape induced by IFN-Is to act in contradictory ways, as one way can be protective, while another could exacerbate the inflammatory environment and cause immune cell dysfunction. These findings support further investigation into the epigenomic signatures and the relationships among IFN-I signaling, chromatin changes, and DNA methylation in the context of HIV infection.

The interplay of metabolic dysregulation, cytokine- and APC-driven suppression, epigenetic reprogramming, and lymphoid tissue remodeling creates a multifaceted impairment of T cell function under chronic Type I IFN exposure. Together, these mechanisms synergize to drive profound T cell exhaustion, characterized by reduced proliferative capacity, diminished cytokine production, and impaired cytotoxicity. Understanding the specific mechanisms by which type I IFNs impair T cell function provides critical insights into potential therapeutic approaches to fine-tune IFN-I signaling and restore immune competence.

## 5. Potential Therapeutic Strategies to Modulate Type I IFN Signaling in HIV Infection

The therapeutic modulation of IFN-I signaling primarily employs monoclonal antibodies targeting the IFNAR or its ligands and IFNAR antagonists specifically designed to block IFN-I binding, thereby suppressing the activation of downstream signaling cascades. This approach has shown significant promise in preclinical models of chronic viral infections [23,144,145]. In HIV infection, blocking IFNARs has been shown to restore the functionality of T cells, decrease inflammatory cytokine production, and enhance viral suppression. Studies in humanized mouse models of HIV infection demonstrated that IFNAR antagonists reduce immune exhaustion markers like PD-1 and improve the cytotoxic capacity of T cells [31,146]. However, blocking IFN-I during acute SIV infection in rhesus macaques resulted in the reduced expression of antiviral genes, increased size of the SIV reservoir, and accelerated CD4 T cell depletion and progression to AIDS [69]. In contrast, treatment with the IFN-1 antagonist in both ART-suppressed and ART-untreated chronically SIV-infected animals resulted in a reduction in IFN-I-mediated inflammation but had no effect on plasma viremia level [147].

Another approach to modulating excessive IFN-I signaling involves targeting the JAK-STAT pathway, which mediates the downstream effects of IFNAR activation. Small-molecule inhibitors of JAKs, such as Ruxolitinib, have shown effectiveness in reducing chronic inflammation and immune activation. In HIV research, in vitro studies have demonstrated that JAK inhibitors can reduce the frequency of cells harboring integrated HIV viral DNA and block the interleukin-15 (IL-15)-induced reactivation of latent HIV [148]. Additionally, a randomized trial indicated that Ruxolitinib significantly reduces immune activation in individuals living with HIV [149]. In the context of HIV, chronic IFN-I signaling not only impairs systemic immune responses but also plays a critical role in maintaining viral reservoirs and contributing to central nervous system (CNS) complications. Recent studies have also explored the potential of JAK inhibitors in mitigating the effects of HIV associated neurocognitive disorders (HANDs). In a severe combined immunodeficiency (SCID) mouse model intracranially injected with HIV-infected human monocyte-derived macrophages (MDMs), baricitinib, a JAK1/2 inhibitor, was shown to cross the blood–brain barrier and reverse cognitive deficits associated with HANDs [150]. By dampening chronic IFN-I signaling, Ruxolitinib treatment alleviated neuroinflammation, improved cognitive deficits, and reduced histopathological damage [151]. Those studies demonstrated that inhibiting the JAK/STAT signaling pathway could be a potential therapeutic strategy to finetuning type I IFN signaling for reducing HIV replication and managing HIV-related neurological complications.

Beyond direct inhibition of the IFN-I signaling pathway, modulating cellular processes that influence upstream innate sensing and downstream immune exhaustion offers an additional layer of therapeutic potential. Emerging evidence indicates that autophagy helps degrade cytosolic mitochondrial DNA (mtDNA) and viral nucleic acid–containing complexes, thereby limiting the chronic activation of cytosolic sensors such as cGAS-STING, which are key drivers of sustained IFN-I production [152,153,154]. In contrast, impaired autophagy leads to increased IFN-I responses [155,156,157]. Our recent study showed that the induction of autophagy with rapamycin reduced ISG expression, alleviated T cell exhaustion, and enhanced CD8⁺ T cell function in the setting of chronic HIV-induced immune dysfunction [156]. Taken together, these findings support a model in which targeting autophagy can fine-tune IFN-I signaling by limiting excessive innate immune activation at its source while simultaneously supporting T cell survival and function. Given its dual benefits, modulating autophagy to fine-tune type I IFN signaling may offer an alternative strategy to improve outcomes in chronic viral infection and immune exhaustion.

Considering IFN-I’s important role in innate immune responses during acute infection, the carefully timed and dosed administration of IFN-I can provide substantial benefits in viral infectious disease. This dichotomous nature of IFN-I signaling underscores the need for strategies that enhance its beneficial effects while minimizing its harmful consequences, particularly in the context of chronic inflammation and immune dysfunction. Recent research has shown that the timing and dosing of IFN-I administration or blocking are critical for IFN-I-based therapies [53,141,158]. During acute viral infections, the short-term administration of type I IFNs can amplify innate and adaptive immune responses, promoting rapid viral clearance in hepatitis C [53,159]. In HIV research, however, this concept has unique challenges due to the virus’s ability to persist and exploit immune activation. Early recombinant IFNα2a treatment in acute SIV infection enhanced antiviral responses, but prolonged use led to IFNα/β desensitization, increasing the SIV reservoir and CD4+ T cell loss [69]. In contrast, the blockade of IFNAR in combination with ART in HIV infected humanized mice facilitated faster viral suppression and reduced viral reservoir [31,146]. These findings underscore the critical role of timing in IFN-induced innate immune responses during acute SIV infection, suggesting that the timing of such responses is a key factor influencing disease progression, a concept likely applicable to various other infections.

Combination therapies involving IFN-I modulation offer a promising strategy for enhancing immune responses in both cancer and viral infections. Pairing IFN-I therapies with immune checkpoint inhibitors, such as anti-PD-1 or anti-CTLA-4 antibodies, has shown potential in reversing T cell exhaustion and improving the efficacy of immunotherapies. In cancer research, blocking IFNAR in combination with immune checkpoint inhibitors has been explored as a way to reinvigorate anti-tumor immunity [160]. Additionally, sustained type I IFN signaling has been identified as a mechanism of resistance to ICB, suggesting that modulating IFNAR activity could potentially overcome this resistance and improve therapeutic outcomes [161].

CAR-T cell therapy represents a promising “kill” component within the “shock and kill” strategy aimed at eradicating latent HIV reservoirs [162,163]. However, during chronic HIV infection, persistent IFN-I signaling and sustained immune activation drive CAR-T cell exhaustion, posing a major barrier to therapeutic success. Our recent study demonstrated that rapamycin, an mTOR inhibitor, can counteract these challenges by enhancing mitochondrial function, reducing chronic inflammation, and preventing CAR-T cell exhaustion [164]. These effects collectively support a more favorable immune environment for CAR-T activity, potentially increasing the efficiency of reservoir clearance. By alleviating IFN-I-driven immune dysfunction, rapamycin may help “recondition” the immune system, enabling more robust CAR-T cell responses in “shock and kill” approaches [156,164].

Targeting pDCs has also emerged as a strategic target for dampening the dysregulated type I IFN response in autoimmune diseases like SLE. Targeting ILT7 with the monoclonal antibody VIB7734 significantly reduced circulating and skin-resident pDCs in patients with cutaneous lupus erythematosus, leading to decreased local IFN-I activity and clinical improvement [165,166], while transient pDC depletion in a lupus-prone mouse model similarly diminished immune cell activation, autoantibody production, and kidney inflammation, underscoring the central role of pDC-driven IFN-I in lupus pathogenesis [167]. Translating this approach to HIV, where pDCs sustain a high level of IFN-I during chronic infection, is compelling. This mechanistic rationale highlights pDCs targeting as a potential adjunctive strategy in chronic HIV infection—by curtailing the chief source of IFN-I, it may be possible to break the cycle of IFN-driven inflammation and immune dysfunction, thereby bolstering host immunity and improving clinical outcomes [6].

The integration of these strategies—precisely timed IFN-I administration, combination therapies with checkpoint inhibitors, and metabolic enhancers—offers a powerful framework for improving immune responses in both virology and cancer. In the context of HIV, leveraging the antiviral properties of type I IFNs while minimizing their chronic effects has the potential to enhance curative strategies [26,168]. Future research will need to refine these approaches further, with an emphasis on understanding how to balance the pro- and anti-inflammatory effects of IFN-I signaling in different stages of infection or disease progression [169]. Another primary challenge lies in balancing efficacy and safety. Patient-specific factors, such as immune status, genetic predisposition, and underlying comorbidities should also be considered. Finetuning the delivery of IFN-I therapies to achieve the right balance between efficacy and safety will require extensive clinical trials and personalized treatment approaches.

## 6. Conclusions

Type I IFN signaling is a critical determinant of immune responses in HIV infection. While its early antiviral effects are indispensable, chronic dysregulation poses significant challenges to T cell immunity. Advances in our understanding of IFN-I signaling dynamics and the development of targeted therapeutic strategies offer hope for enhancing immune function and achieving better outcomes for individuals living with HIV. The future of IFN-I modulation lies in overcoming these challenges through innovative research and clinical advancements. Future research should focus on developing personalized treatment protocols that effectively balance the therapeutic benefits and potential risks of IFN modulation. In addition, developing robust biomarkers for patient stratification and treatment monitoring as well as unraveling the complex interactions between IFN-I and latent reservoirs are critical steps forward. Moreover, identifying the key ISGs that regulate type I IFN signaling during chronic infection could provide novel tools to fine-tune IFN responses at different stages of HIV infection. By addressing these challenges, IFN-I-based therapies have the potential to transform treatment paradigms for viral infections, bringing us closer to achieving long-lasting immune restoration and a functional cure for HIV infection.

## Data Availability

No specific data were generated to support reported results.

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
