# Peer review of "Finetuning Type I Interferon Signaling to Enhance T Cell Immunity in HIV Infection"

_viruses, 2025, doi:10.3390/v17060774_

Round 1
Reviewer 1 Report
Comments and Suggestions for Authors
In this manuscript, the authors comprehensively review the dual role of type I interferons (IFN-I) in HIV infection, highlighting their protective and pathogenic effects during acute and chronic phases. The review also summarizes recent advances in understanding the mechanisms underlying IFN-I-mediated pathogenesis in HIV and explores potential therapeutic strategies targeting IFN-I signaling.
Major Comments:
Section Title Revision:
- In the section “Mechanisms of Immune Response Impairment by Type I IFNs,” the authors primarily focus on direct IFN-I effects on T-cell function. However, indirect mechanisms—such as modulation of antigen-presenting cells (APCs) or lymphoid tissue architecture—are also relevant. While some of these aspects are mentioned elsewhere, integrating them into this section would provide a more cohesive discussion of IFN-I-mediated immune dysregulation.
- The section titled 'Therapeutic Strategies to Modulate Type I IFN Signaling' should be revised to 'Potential Therapeutic Strategies to Modulate Type I IFN Signaling in HIV Infection' to more accurately reflect the experimental or preclinical nature of these approaches, as they have not yet been implemented in clinical practice.
- The authors should incorporate literature on plasmacytoid dendritic cell (pDC) depletion or inhibition as an alternative approach to IFN-I modulation, particularly given its relevance in autoimmune diseases. This would provide a more comprehensive perspective on potential therapeutic interventions.
Minor Comments:
- The abbreviation of type I interferons as “IFN-α/β” is misleading, as IFN-I encompasses additional subtypes beyond IFN-α and IFN-β. The authors should consistently use “IFN-I” after its initial definition.
- On Page 3, “plasmacytoid dendritic cells (pDCs)” repeats the full name unnecessarily, as it has already been introduced and abbreviated on Page 2.
- While “type I interferon (IFN-I)” is correctly abbreviated in the first paragraph, subsequent paragraphs inconsistently reintroduce both the full term and abbreviation. The authors should maintain uniform usage of “IFN-I” after the first instance.
- On Page 2, the sentence: “During the acute phase of HIV infection, type I IFNs are rapidly produced by infected cells and plasmacytoid dendritic cells (pDCs) upon sensing viral RNA through PRRs such as TLR7 and RLRs, such as cytosolic sensors (e.g., cGAS-STING).” requires revision, as RLRs (RIG-I-like receptors) are themselves cytosolic sensors. Consider rewording for clarity as the following “...through PRRs, including endosomal TLR7 and cytosolic sensors (e.g., RLRs and cGAS).”
- On Page 3, the citation formatting in the 2nd and 3rd paragraphs does not match the style used elsewhere in the manuscript. Please ensure uniformity throughout.
Author Response
In this manuscript, the authors comprehensively review the dual role of type I interferons (IFN-I) in HIV infection, highlighting their protective and pathogenic effects during acute and chronic phases. The review also summarizes recent advances in understanding the mechanisms underlying IFN-I-mediated pathogenesis in HIV and explores potential therapeutic strategies targeting IFN-I signaling.
We would like to thank the reviewer’s positive and constructive comments.
Major Comments:
Section Title Revision:
1. In the section “Mechanisms of Immune Response Impairment by Type I IFNs,” the authors primarily focus on direct IFN-I effects on T-cell function. However, indirect mechanisms—such as modulation of antigen-presenting cells (APCs) or lymphoid tissue architecture—are also relevant. While some of these aspects are mentioned elsewhere, integrating them into this section would provide a more cohesive discussion of IFN-I-mediated immune dysregulation.
Response: We appreciate and fully agree with the reviewer’s suggestion. We have revised the section to incorporate the modulation of antigen-presenting cells (APCs) and disruption of lymphoid tissue architecture as key indirect mechanisms contributing to IFN-I–mediated immune impairment, as recommended.
2. The section titled 'Therapeutic Strategies to Modulate Type I IFN Signaling' should be revised to 'Potential Therapeutic Strategies to Modulate Type I IFN Signaling in HIV Infection' to more accurately reflect the experimental or preclinical nature of these approaches, as they have not yet been implemented in clinical practice.
Response: We have revised this section as recommended.
3. The authors should incorporate literature on plasmacytoid dendritic cell (pDC) depletion or inhibition as an alternative approach to IFN-I modulation, particularly given its relevance in autoimmune diseases. This would provide a more comprehensive perspective on potential therapeutic interventions.
Response: We appreciate and fully agree with the reviewer’s suggestion. We have added a standalone paragraph summarizing the therapeutic potential of plasmacytoid dendritic cell (pDC) depletion or inhibition as a strategy to modulate IFN-I signaling.
Minor Comments:
1. The abbreviation of type I interferons as “IFN-α/β” is misleading, as IFN-I encompasses additional subtypes beyond IFN-α and IFN-β. The authors should consistently use “IFN-I” after its initial definition.
Response: IFN-α/β was changed to IFN-I to keep consistent as recommended.
2. On Page 3, “plasmacytoid dendritic cells (pDCs)” repeats the full name unnecessarily, as it has already been introduced and abbreviated on Page 2.
Response: Repeated abbreviation has been deleted.
3. While “type I interferon (IFN-I)” is correctly abbreviated in the first paragraph, subsequent paragraphs inconsistently reintroduce both the full term and abbreviation. The authors should maintain uniform usage of “IFN-I” after the first instance.
Response: Repeated abbreviation has been deleted.
On Page 2, the sentence: “During the acute phase of HIV infection, type I IFNs are rapidly produced by infected cells and plasmacytoid dendritic cells (pDCs) upon sensing viral RNA through PRRs such as TLR7 and RLRs, such as cytosolic sensors (e.g., cGAS-STING).” requires revision, as RLRs (RIG-I-like receptors) are themselves cytosolic sensors. Consider rewording for clarity as the following “...through PRRs, including endosomal TLR7 and cytosolic sensors (e.g., RLRs and cGAS).”
Response: Thank you for the comments. The sentence was revised as recommended.
4. On Page 3, the citation formatting in the 2nd and 3rd paragraphs does not match the style used elsewhere in the manuscript. Please ensure uniformity throughout.
Response: Thank you for the comment. The references have been reformatted to ensure consistency throughout the manuscript.
Reviewer 2 Report
Comments and Suggestions for Authors
Review article by Wenli Mu, Nandita Kedia and Anjie Zhen “Finetuning Type I Interferon Signaling to Enhance T Cell Immunity in HIV Infection”
There is an ongoing debate about the dual role of the interferon (IFN) system in acute and chronic viral infections and other pathological conditions [e.g., Canar J, Darling K, Dadey R, Gamero AM. The duality of STAT2 mediated type I interferon signaling in the tumor microenvironment and chemoresistance. Cytokine. 2023 ;161:156081. doi: 10.1016/j.cyto.2022.156081; Lee AJ, Ashkar AA. The Dual Nature of Type I and Type II Interferons. Front Immunol. 2018 Sep 11;9:2061. doi: 10.3389/fimmu.2018.02061. PMID: 30254639; PMCID: PMC6141705]. IFN type 1 (IFN-I) is key to controlling viral replication and triggering innate immune responses during acute viral infection. However, prolonged activation of the IFN system inhibits the antiviral response and suppresses innate immune responses, leading to chronic infection. The authors of this review analyze possible mechanisms of dysregulation of IFN-I production and action in chronic HIV-1 infection and substantiate the need to correct IFN-I signaling during antiretroviral therapy. A defining feature of chronic HIV infection is the preservation of IFN-I responses, characterized by persistent ISG expression and prolonged immune activation. This contributes to immune exhaustion, characterized by decreased T cell proliferation, impaired cytokine production, and decreased cytotoxic activity. From this perspective, the work is relevant and timely.
The review is written in clear language and provides compelling evidence for the need to develop strategies that enhance the beneficial effects of IFN-I while minimizing its harmful effects, especially in the context of chronic inflammation and immune dysfunction. The authors conclude that “the timing and dosage of IFN-I administration or blockade are critical for IFN-I-based therapy.” Furthermore, they conclude promisingly that “combination of IFN-I therapy with immune checkpoint inhibitors, such as anti-PD-1 or anti-CTLA-4 antibodies, shows potential in reversing T cell exhaustion and enhancing the efficacy of immunotherapy.” The conclusions drawn by the authors follow logically from the problem under discussion, and outline the necessary steps for future development of reliable biomarkers for patient stratification and promotion of personalized treatment protocols for AIDS patients. The cited sources are mostly confirmed by recent publications and are relevant.
The review may be published in the journal Viruses.
Author Response
Response: We appreciate the reviewer’s thorough and positive evaluation and recommendation for publication.
Reviewer 3 Report
Comments and Suggestions for Authors
Mu et al summarize recent advances in our understanding of chronic inflammatory signaling and immune exhaustion in patients living with HIV. The authors discuss the antiviral benefits of IFN signaling during the early/acute phase of infection and the deleterious consequences of prolonged IFN signaling during late-phase infection even in the presence of ART. Modulating type I IFN responses has emerged as a novel therapeutic strategy to restore immune function, enhance T cell efficiency, and improve clinical outcomes in patients living with HIV. Overall, this review is clear and concise and I have only minor comments.
Minor:
- Several missing spaces between acronyms and parenthesis in the second intro paragraph. First reference to SIV is the same.
- References didn’t format properly in the paragraph immediately before the section titled “chronic dysregulation of type I IFN signaling.”
Author Response
Mu et al summarize recent advances in our understanding of chronic inflammatory signaling and immune exhaustion in patients living with HIV. The authors discuss the antiviral benefits of IFN signaling during the early/acute phase of infection and the deleterious consequences of prolonged IFN signaling during late-phase infection even in the presence of ART. Modulating type I IFN responses has emerged as a novel therapeutic strategy to restore immune function, enhance T cell efficiency, and improve clinical outcomes in patients living with HIV. Overall, this review is clear and concise and I have only minor comments.
We would like to thank the reviewer’s positive and constructive comments.
Minor Comments:
1. Several missing spaces between acronyms and parenthesis in the second intro paragraph. First reference to SIV is the same.
Response: Thank you for the comment. The missing spaces between acronyms and parentheses have been corrected as recommended.
2. References didn’t format properly in the paragraph immediately before the section titled “chronic dysregulation of type I IFN signaling.”
Response: Thank you for the comment. The references have been reformatted to ensure consistency throughout the manuscript.
Round 2
Reviewer 1 Report
Comments and Suggestions for Authors
My previous comments have been well amended, and I have no further comments.
Reviewer 3 Report
Comments and Suggestions for Authors
No further comments or concerns